# RePrompt: Prompt Engineering for Large Language Model Agents through Reflections

## Abstract

In this past year, large language models (LLMs) have had remarkable success in domains outside the traditional natural language processing, and people are starting to explore the usage of LLMs in more general and close to application domains like code generation, travel planning, and robot controls. Connecting these LLMs with great capacity and external tools, people are building the so-called LLM agents, which are supposed to help people do all kinds of work in everyday life. In all these domains, the prompt to the LLMs has been shown to make a big difference in what the LLM would generate and thus affect the performance of the LLM agents. Therefore, automatic prompt engineering (APE) has become an important question for many researchers and users of LLMs. However, previous works in APE all rely on a final checker to evaluate the performance of the given prompt, which is hard to meet in the case of LLM agents where intermediate feedback is easier to get, and the final evaluation could be expensive, inaccurate, or even missing. In this paper, we propose a novel method, RePrompt, which does a "gradient descent"-like approach to optimize the step-by-step instructions in the prompts given to LLM agents, based on the chat history obtained from interactions and reflections with LLM agents. By leveraging intermediate feedback, RePrompt can optimize the prompt without the need for a final solution checker. We have used experiments in PDDL generation and travel planning to show that our method could generally improve the performance for different reasoning tasks.

## 1 Introduction

Large language models (LLMs) have won significant success since the release of ChatGPT (OpenAI, 2022). In addition to traditional natural language tasks like summarization and sentiment analysis, LLMs have been shown to be effective in many domains that are closer to applications like code generation (Chen et al., 2023; Roziere et al., 2023), human-computer interaction (Li et al., 2023) and math problem solving (Wei et al., 2022; Yu et al., 2024). While pure LLMs are limited in their reasoning capability (Sun et al., 2023; Valmeekam et al., 2023; Chen et al., 2024a), researchers have introduced tool-use to LLMs and built integrated systems, namely LLM agents, to enable the possibility of using LLM in even more general domains like robot controls (Wang et al., 2023a) and autonomous driving (Mao* et al., 2023).

Behind all of these successes, prompts are playing an important role. It has been shown that different prompts could lead to completely different success rates (Wei et al., 2022), and hence prompt engineering is often needed for each specific task. Because prompt engineering is difficult and time-consuming, automatic prompt engineering (APE) has emerged as a strategy, where the LLMs themselves write in the clear language they prefer (Zhou et al., 2023). With a limited number of trials, APE can efficiently converge to a robust prompt that performs better than a simple prompt on traditional natural language processing tasks (Zhou et al., 2023; Zhang et al., 2023).

However, in complex reasoning tasks, APE is still under-studied and most users still use primitive prompts or carefully hand-crafted prompts in their LLM. In complex reasoning tasks, also known as LLM agents, the performance of a prompt—measured by the solution's success rate—can be much more costly compared to question-answering (QA) tasks, which previous APE methods mainly focus on. The exact evaluation of the task could either require human experts to spend a long-time to evaluate, or when collecting chat history online, users will try to provide feedback in the middle

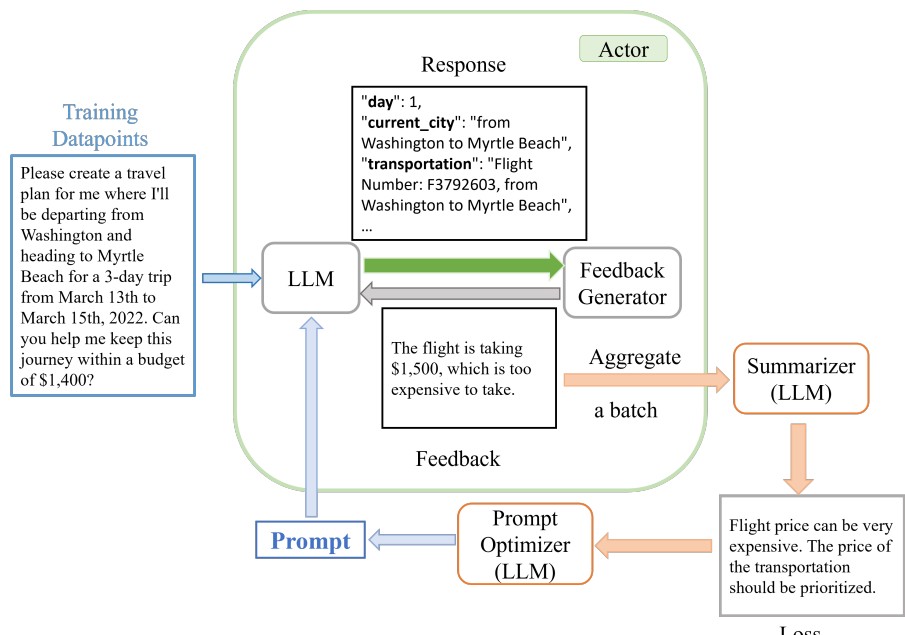

Figure 1: The workflow of our method REPROMPT.
.

to help the LLMs refine the answers, but will leave without telling the LLM-agent system whether they are leaving because they have get the desired response, or they are leaving because they think LLM-agents are consistently failing. Without cheap prompt performance evaluators, existing APE methods cannot be applied to the scenarios.

In this paper, we focus on a scenario where there is a specific reasoning task one wants to use LLMs for, with examples like people already choosing to use a specific OpenAI GPTs tool in ChatGPT (OpenAI, 2023b) to plan their travel or help in writing a code. In these domains, LLMs typically use a Chain-of-Thought (CoT) prompt with interactive procedures like REACT (Yao et al., 2023b) and REFLEXION (Shinn et al., 2023) to improve their performances. We propose a novel automatic prompt engineering method called REPROMPT, which takes the common practices of using CoT and REACT into consideration and uses the dialogue history from these results as the information for each prompt update. By summarizing the dialogue history and then analyzing how to improve the prompt sentence by sentence, we can successfully optimize the prompt, based on past history while not overfitting to corner cases. We further show how our proposed method could be seen as a standard fine-tuning procedure in ML where the changeable parameter (or input) is the initial prompt. An overview of our proposed framework is shown in Figure 1.

We use experiments on Planning Domain Definition Language (PDDL) generation (Guan et al., 2023) and travel planning (Xie et al., 2024) to show that our methods can achieve a higher first-round success rate. These scenarios demonstrate the effectiveness of our approach when feedback is either expensive but accurate, or cheap but less accurate and informative.

In conclusion, our contributions are:

1. Propose to use "gradient-based"-like prompt optimization in LLM agents.

2. Propose a summarization-based method to give specific instruction on how the current prompt could be further improved, and propose a novel guideline to optimize the prompt in a step-by-step format.

3. Our proposed method does not require a solution checker, and can be used in LLM-agents scenarios where such a checker is not available.

4. Using updated prompts, improve the results in multiple LLM agent benchmarks without fine-tuning LLM models on them.

## 2 RELATED WORKS

Our work lies in the intersection of prompt optimization and LLM for reasoning.

In prompt optimization, many works proposed to optimize the prompt using differentiable tuning on soft prompts (Lester et al., 2021; Qin & Eisner, 2021), train auxiliary models as the optimizer (Hao et al., 2023; Deng et al., 2022; Zhou et al., 2023), or directly train the prompter themselves (Wang et al., 2023b). This line of work requires access to the model weights of the language models and is not generally applicable in the current era of using LLMs like GPT-4 (OpenAI, 2023a) and Claude-3 (Anthropic, 2024) through APIs. Another line of work chooses to use machine learning models to provide the approximated guidance on what prompt is better, either by using reinforcement learning (Shin et al., 2020; Zhang et al., 2023; Chen et al., 2024b) or doing discrete manipulation with LLM feedbacks (Guo et al., 2023). There are also some other works in prompt optimization that propose a relatively general solution, like using beam search or Monte Carlo tree search as the "gradient descent" optimizer (Pryzant et al., 2023; Tang et al., 2024; Wang et al., 2024). Our work is very close to the second group of work, and can be seen as a generalization from the current methods to domains that are more on reasoning domains. However, in our settings, we do not necessarily require ground-truth feedback that checks the general performance of the current prompt, which is more aligned with the potential applications of prompting for LLM agents in scenarios like GPTs OpenAI (2023b).

In LLM for reasoning, a key challenge identified by researchers is how to correctly use prompts to guide the LLM to generate useful auxiliary output that leads to a good final solution. Chain-of-Thought (CoT) (Wei et al., 2022) is the most commonly used prompt that can improve the performance of LLMs that consists of simply adding a fixed sentence of "Let's think step by step.". Later on, Tree-of-thought (Yao et al., 2023a) and Graph of Thoughts (Besta et al., 2024) are also proposed as an extension from the simple line-based architecture of auxiliary output to tree and graph-structured output. Orthogonally, researchers have also found that utilizing the interaction capability of LLMs could also improve their performance. Yao et al. (2023b) proposed REACT by letting the LLMs list some thoughts before proposing the actual actions. REFLEXION (Shinn et al., 2023) prompts an LLM agent to reflect on the actions, and save the reflections in the memory to improve the efficiency. There are extensive amounts of works like self-refine (Madaan et al., 2023), RCI (Kim et al., 2023), and self-debugging (Chen et al., 2023) that use similar ideas of providing feedback as guidance in the reasoning-related task and strategically adapt the idea to fit the needs of specific domains. Our work considers this popular and useful workflow of iterating with feedback before finalizing the answer and optimizes the prompt based on the interaction history provided. This approach helps to avoid early-stage errors and, more importantly, reduces the randomness observed in methods like ReAct and Reflexion, where specific scenarios may not receive adequate feedback. By summarizing common issues, such as budget balancing, REPROMPT ensures these are addressed in the prompt, allowing ReAct and Reflexion to focus on case-specific issues, like unexpected price spikes for hotels in certain cities on particular days.

## 3 METHODS

### 3.1 PRELIMINARY ON LLM AGENTS

In this paper, we consider the problem of LLM agents for reasoning tasks. Because LLM might not be correct in the first shot, in certain scenarios where feedback is still possible, prior approaches have proposed allowing a few interactions before the final answer is given by the LLM (Yao et al., 2023b; Shinn et al., 2023). In these cases, users provide the LLMs with some error information and let the LLMs try again with the additional information. This information does not necessarily provide any hint about the final solution but is an error message about why the current solution is not correct. For example, this information could be a typical Python runtime error message in code generation tasks. How to use specific prompts with error messages would mainly depend on different tasks. For example, in the widely used REACT (Yao et al., 2023b), LLMs are required to provide thoughts on the current results before doing the next round, and these thoughts are often not a concrete error but may also include a short analysis that reaches a conclusion that the current action is good. However, due to the difficulty of LLM-agents tasks, checking how good a given prompt is could be expensive and not able to be used regularly in the optimization process. In the closest application available

right now, the GPTs, users normally leave without telling LLM whether the current solution is good enough or it is too far away, and the user does not want to spend any extra effort.

For LLM agents to focus on reasoning, the Chain-of-Thoughts (CoT) Wei et al. (2022) is one of the most popular methods used in LLMs. By adding a simple sentence of "Let's do it step-by-step," LLMs will automatically consider outputting auxiliary steps before generating the final answer. Similar to how writing down the calculation procedure helps a high-school student solve math problems, the auxiliary steps are shown to be effectively helping the LLMs get a higher success rate in solving reasoning questions. From task to task, the auxiliary steps could differ significantly, and the exact steps needed in a task are called planning in LLM agents. A common practice is to let the LLM first generate these steps and then further choose whether each step should be executed by another LLM or external tools.

### 3.2 REPROMPT

#### 3.2.1 OVERVIEW

Specifically, in LLM agents for reasoning tasks, we consider the task planning part of LLM agents with prompt optimization, where the tasks of the agents are known ahead of time, and the final solution checker is not available due to its cost. A typical example of this scenario is the different GPTs in OpenAI (OpenAI, 2023b).

As shown in Fig. 1, our method, REPROMPT, is a prompt optimizer that is based on the interaction-based action generate process. Our method is similar to a machine learning training loop, which iterates between getting the output based on the current parameter, calculating the loss based on the output, and optimizing the parameter based on the loss. But in our case, the parameters to be trained are the prompts going to be fed into the model, the model forward pass is replaced by the complete interaction-based action generate process, which include the feedback information generator, the loss and optimizer are both LLMs instead of numerical calculation for the distance and the gradients.

Given a specific small dataset of reasoning tasks used for training, we first let the LLMs generate their responses using the current prompt. This process needs to include some interaction schemes with some kind of feedback provider like REACT or REFLEXION , but we do not put any constraint on how this part should be done, and how accurate the feedback is. We call this process the act loop.

We then wait until a complete batch of chat histories has been collected, at which point we input the entire batch into a large language model, which we call the summarizer, to summarize the primary focus point. This focus point might be a recurring issue that frequently prolongs iterations or specific suggestions (or "thoughts" in the case of REACT) that have proven effective in producing high-quality responses. Typically, the essential information is already present in the chat history and does not require further analysis or summarization. Our summarizer is designed to capture key insights across various scenarios, omitting scenario-specific details and recommendations while avoiding overly broad summaries that would demand additional reasoning steps. Due to space constraints, we provide this prompt for this summarizer in the appendix.

Then, with this summarized typical error, we use another LLM , the Prompt Optimizer, to update the actual prompt. We ask this optimizer LLM to follow the following update rules:

1. The improvement should focus on the common prompt part rather than scenario-specific prompts that change from data point to data point. For example, in the task of formulation generation for PDDL, one of the most commonly appearing suggestions is to provide more information on what specific domain the LLM is trying to generate and more detailed background knowledge on what the domain is about. However, because we want to get a general enough prompt to solve all the PDDL formulation generation, such an update should not happen.

2. The improvement should prefer to identify whether the specific problem does occur in the given scenario. For example, suppose there is a certain budget one wants the solution provided by LLMs to satisfy, and the previous history shows that this budget constraint could be one of the main problems that lead to a wrong solution. In that case, it should first approximate the cost for a typical plan. If it breaks the constraint, it could prioritize the budget constraint when getting a solution; otherwise, it should ignore this problem.

Based on the above rules, we ask the LLM to the following steps:

1. Propose a few potential solutions to the problem.

2. Analyze the solutions one by one to see which one meets the rules better.

3. Choose the single solution that is the best. Unlike APE and some of the following work (Zhou et al., 2023; Deng et al., 2022), we do not ask the LLM to give a concrete number as the value of the sentence.

4. Analyze the original steps in the original prompts, check whether the chosen solution should be inserted before the current step or the solution is a more concrete detail on the step, and the prompt on the current step should be replaced by the solution. If it is the step, add the prompt here.

5. Output the final prompt that combines the original prompt and the updated prompt.

To initialize the training process, we start with the original prompt. Although not all original prompts include detailed step-by-step instructions that our optimizer can leverage, we introduce an auxiliary checker (excluded from the main workflow for simplicity) to convert prompts into a structured, step-by-step format when needed. Specifically, we first use a language model to assess whether the current prompt already contains such instructions. If it does not, we manually append a standardized sequence of steps to the prompt, placing it just before any examples. This sequence comprises two primary steps: a brief problem analysis followed by the solution. This structure is functionally similar to a single "Chain of Thought" (CoT) prompt (Wei et al., 2022), "let's think step-by-step" in most reasoning tasks by encouraging essential analysis without introducing domain-specific knowledge, this approach maintains generality across tasks and domains.

We integrate the specific rules and procedural steps mentioned above into our optimizer prompt, detailed in the appendix due to page limit. To assist the optimizer in handling common challenges, we have pre-encoded several frequently used solutions directly within the optimizer prompt. While these solutions could be discovered over multiple iterations, providing them upfront minimizes the number of iterations required for prompt refinement, accelerating the optimization process.

After the steps, we will get an updated prompt, and we can continue to do more iterations, which can also be seen as more epochs as an analogy to training ML models, until the prompt has converged and does not change with more iterations. This converged prompt will help improve the generated result in the first round , and also help assure common problems that can be fixed by the feedback generator to be resolved as early as possible. During test time, we only need to use the converged updated prompt and test it on the new test set. During the test, we do not require the exact same process to generate the response, e.g., the feedback generator can be removed completely from the Act procedure if it is quite expensive.

Note that in the optimization process, REPROMPT only changes the step-by-step instruction phase rather than any other problem description or format requirement. This brings us to two possible formats of prompts that the algorithm will end up with:

1. If the current prompt is ReAct-like, which already includes a step-by-step instruction that gives a specific step, like Thought in REACT, to include all the potential analysis, our prompt will converge to always update this thinking step by adding more and more hints on what to do to it. Our method becomes an algorithm that gives a more specific hint on what part the analysis should focus on compared to other prompt engineering work that introduces hints dynamically.

2. If the prompt is step-by-step, like solving a math or logical problem, then our algorithm is very likely to add more and more steps to the procedure generated by the planning. This will lead to a more concrete direction to try and focus on in the planning process, will guide the LLM in getting to the correct final answer, and will serve as the planner that breaks down the high-level task.

While we understand that in-context learning is very important for reasoning, we found it extremely challenging to update the examples to follow our instructions step-by-step perfectly all the time. And therefore, we choose not to change the examples at all. In most cases, the examples serve as a hint to the LLM on the output format and the related capabilities rather than a concrete guide on

| | # of Incorrect Actions | | | Total # of Errors | | |
|---|---|---|---|---|---|---|
| | TyreWorld | Logistics | Household | TyreWorld | Logistics | Household |
| Dataset Size | 13 | 6 | 22 | INF | INF | INF |
| Guan et al. (2023) | 3 | 1 | 19 | 6 | 1 | 52 |
| REPROMPT | 3 | **0** | **12** | **4** | **0** | **23** |

Table 1: The results on generating PDDL instances correctly without any additional domain expert help. The number of actions is the number of tests provided to the LLMs, and each action can have as many errors as it wants, annotated by human experts. Our method, REPROMPT, is trained for only 1-epoch with only the annotations used to evaluate the original results, and without additional annotation from human experts in training.

how to follow the step-by-step instructions, and we currently do not see any empirical drawback by not updating them.

# 4 EXPERIMENTS

## 4.1 EXPERIMENT SETTINGS

In the experiments, the stopping criteria and other hyperparameter settings for each domain will be the same as in the original environments. Without specifications, we use a temperature $= 0$, and a seed of $42$. All results are tested on GPT-4-turbo-1106-preview, given its strong general capability (Chiang et al., 2024). To help reproducibility, we provide all the optimized prompts generated by REPROMPT in the appendix.

To test the capability of our algorithm in different scenarios, we choose two environments, PDDL generation (Guan et al., 2023), and Travel Planner (Xie et al., 2024). The two tasks are selected because their feedback generator is already included in the paper, and the reality is that both datasets are hard to conduct iterations with feedback generators. The PDDL generation task provides accurate but expensive feedback and challenges the exact translation capability of the LLMs, which is necessary for LLMs to be further able to write correct code. The TravelPlanner environment, on the other hand, provides cheap but not accurate feedback through Reflexion without knowing ground-truth information. Travel Planner also provides tools to be used to query the cost information in the database and challenges the reasoning capability by asking for direct generating solutions. Furthermore, in TravelPlanner, we are testing REPROMPT with REFLEXION, which further includes the thought-action-observation steps rather than a standard step-by-step instruction in the PDDL where each step is an intermediate step that could be helpful for guiding the generation for final results. Given the different types of feedback, the purpose of our REPROMPT also change: in PDDL, our REPROMPT serves to improve the generation performance without any iteration between the LLM actor and the feedback generator that is used to reduce the cost of generating feedback, while in TravelPlanner, REPROMPT is used to help guide the LLM to take all the important steps of concern in all scenarios and reduce the potential failures.

## 4.2 PDDL GENERATION

We first test the Planning Domain Definition Language (PDDL) generation task (Guan et al., 2023). Given a natural language description of a PDDL instance, the job is to define the precondition and the effects of the actions in PDDL. Specifically, we consider the very first step of constructing the model and do not further consider the later correction phase and PDDL translation phase. [1] After generating the preconditions and effects of the actions, human domain experts are introduced to check whether the generation is correct. In this paper, we not only have human domain experts but also use another LLM as a separate checker to verify whether any of the errors that appear in the results given by the prompt in the original paper (Guan et al., 2023), which are released together with the code of it, have also appeared in our result.

---

[1] At the time of submission of our paper, the evaluation phase is missing in the official Github repository, and we are not able to compare the success rate in those phases in a fair manner.

In this experiment, we use the generated result and the annotation from human experts of the prompt from the previous paper Guan et al. (2023) in "Tyreworld" as the chat history used in the REPROMPT training set, and update it for one epoch to get the updated prompt. Here, because the feedback is provided by domain experts, it is accurate but expensive, so multiple rounds of iterations are not feasible, and so we choose to only train REPROMPT for one epoch and greatly reduce the need for extra annotations. As shown in Table. 1, the prompt we get from REPROMPT not only outperforms in the set that we are training on, i.e., the Tyreworld domain, but also generalizes to other related domains and improves the success rate there. Interestingly, we found that after changing the prompt, the prompt does not introduce any new errors, i.e., the errors the new prompt made are a subset of the errors made by the prompt in the original paper. With this subset of errors, fewer domain experts will be needed to give annotations, and make the whole PDDL translation process much faster.

Among the remaining errors, some of them are caused by missing common knowledge that is addressed in the description. For example, in the action "Empty a Vacuum Cleaner", the action description includes one sentence saying, " The trash can should be opened if it's openable." This is common knowledge, and it has no additional information. However, in the current context, this sentence indicates one precondition. However, the PDDL precondition generated by the LLM has been omitted, which leads to one error. Similar problems also happen a few times and contribute to a significant number of errors that could be classified together.

## 4.3 TRAVEL PLANNER

Next, we test on the sole-planning setting in TravelPlanner benchmark (Xie et al., 2024). [2] In this benchmark, the LLMs are required to provide a concrete day-to-day plan, including where they should stay, eat, and how they should travel, and satisfying both commonsense constraints like reasonable city routes and budget constraints. While there are some breakdowns of what specific kind of constraints the plan does not satisfy, the primary metric that is used for comparing different methods is the final pass rate. It needs to be addressed that in this benchmark, the evaluation is done after the act loop is done, separately with a ground-truth checker rather than directly to the feedback loop in REFLEXION, and thus, the feedback in the chat history used by REPROMPT does not actually involve any human interference or oracles on what is the correct answer and what is the list of constraints. This allows us to train on a subset of the test set without worrying about leaking any extra oracle information to the model. Because of this, we choose to report results on the validation set of 180 data points instead of results on the larger test set in order to save API costs for our model. [3] And because REPROMPT is based on further collecting data, we choose a subset of 10 data points in the validation set as our training dataset. We report the same group of metrics defined in the original paper Xie et al. (2024).

As shown in Table. 2, the prompt generated by 5 epochs of training of REPROMPT is better than the pure REFLEXION Shinn et al. (2023) result in the final pass rate, and also outperform PromptAgent Wang et al. (2024) as an example of previous automatic prompt engineering work that was primarily designed for single-round question-answering task [4]. In the optimization procedure, unlike the PDDL environment, the optimized prompt after one epoch does not show any benefit. This is because, as we discussed earlier, the prompt after the first epoch will only include one additional suggestion, which is about looking into the budget constraint in this case. No matter what this round of updates is, it is something summarized from the thoughts provided by the REACT scheme and something the iteration loop of generating a final plan has often already noticed and addressed. The optimized prompt after 5 epochs helps LLMs perform better in both the data set we used for training and the other data not included in the optimization process. This shows the generalizability of the

---

[2] We have slightly changed some code to recognize some specific format in the automatic checking procedure. We guarantee that all the changes are for the outputs that have previously thrown an error in the checker program, but not the ones that have already been automatically rejected. While this does not necessarily lead to a higher delivery rate, the delivery rate may show a small improvement compared to the original code, and it will not affect other metrics.

[3] With the 180 validation data points tested by REFLEXION strategy on sole-planning, the cost to finish one generation is $300-800 depending on what exact reflection is provided to the model. The full test set includes 1000 data points, and the cost will grow around linearly in the number of data points tested.

[4] Because their design was not suitable for Travelplanner, we have made necessary adaptations to make it testable in this case. We provided more details in the appendix.

| | Delivery Rate | Commonsense Pass Rate | | Hard Constraint Pass Rate | | Final Pass Rate | Train Set Final Pass Rate |
|---|---|---|---|---|---|---|---|
| | | Micro | Macro | Micro | Macro | | |
| Reflexion | 76.67% | 56.39% | 3.89% | 37.39% | **33.89%** | 2.78% | 1/10 |
| PromptAgent | 94.44% | 56.39% | 3.89% | 32.61% | 30.22% | 2.11% | **2/10** |
| REPROMPT (1-epoch) | 89.44% | 64.03% | 3.89% | 35.0% | 32.78% | 2.78% | 1/10 |
| REPROMPT (5-epochs) | **99.44%** | **80.00%** | **6.11%** | **48.81%** | 25.56% | **3.89%** | **2/10** |

Table 2: Results on Travel Planner Benchmark. The best results are marked in bold. The delivery rate, commonsense pass rate, and hard constraint pass rate overall contribute to the final pass rate, which is the main metric in this table. Because we are "training" on part of the data, but not using any additional ground-truth information, we split the results into overall final pass rate and training set final pass rate. All the metrics in the table are better when the numbers are larger.

| # Training Samples | Final Pass Rate |
|---|---|
| 1 | 1.22 |
| 2 | 2.44 |
| 5 | 3.44 |
| 10 | 3.89 |
| 25 | 2.00 |

Table 3: Training Samples and Final Pass Rate

prompt we get through the process. We observe that our baseline, PromptAgent, tends to change an extensive amount on the original prompt, and became even worse on the final test.

Further looking into where we have gotten our improvement on the final pass rate, we found that our method helps in improving the macro commonsense pass rate (shown in Table 2), which is the major bottleneck in the current stage of using LLM for travel planning. Further breakdown, we found that the prompt has significantly improved the pass rate on the so-called "reasonable city route" (not shown in Table 2, but one of the constraints tested by TravelPlanner (Xie et al., 2024)). This common sense is to ensure "Changes in cities during the trip must be reasonable" (Xie et al., 2024). It is common sense that the first round result without interaction can mostly be satisfied but easily forgotten after the interactions with feedback providers. Thus, this is one of the common sense constraints that LLMs can detect themselves in the process, and it is sometimes, but not often, covered in the thoughts provided in the feedback loop. Our REPROMPT has caught and included this constraint in the prompt, so most iterations will have covered it. We believe this is one example that our algorithm has addressed the challenge of "Agents struggle to align their actions with their reasoning." mentioned in the Travelplanner paper (Xie et al., 2024). However, for our baseline PromptAgent, the extensive changes have made the LLMs fail to follow the desired format and lead to more useless steps and more failure.

To our surprise, we found that our method does not help stop "Agents producing hallucinatory answers due to information confusion.", and our agents still output the wrong flight number that is used for the wrong leg of the flight, and output the same flight number for both departure and arrival flight. While this theoretically could be addressed by the feedback provider, it seems that such feedback is never provided and addressed in the loop, and our REPROMPT training loop cannot catch such an error either. We are looking forward to further investigating how our REPROMPT framework can serve as a further level of checker on simple hallucination errors. In which, one possibility is to combine scenario-specific information with chat history when we calculate the "loss".

## 4.4 ABLATION STUDY

Next, we provide an ablation study on the sampling size and the number of iterations. To fix the total budget spent on training the same, we fix a multiplication of the number of training samples times training epoch to 50. So, with more training samples, one will have fewer training epochs, which might lead to a prompt before it's convergence. As a special case, when training samples are equal to 1, our method is a vanilla prompt optimizer without a summarizer that is done per-scenario. We

show our results in Table. 3. we found that a proper number of samples that balance the diversity with the number of epochs, i.e., 5 or 10 in training samples, could be helpful to REPROMPT. This matches our intuition by matching RePrompt as gradient descent.

# 5 ERROR ANALYSIS

In our experiments, our automatic prompt optimization process does not provide any guarantees of successfully generating a better prompt. In this section, we briefly enumerate a few errors that often occur and describe our methods to address them. We believe that these additional methods are neither necessary nor generally applicable to all domains, and we choose to leave them here just as an ad-hoc solution. We believe that all of these ad-hoc solutions can be automatically removed when the instruction-following capability of LLMs is further improved.

## 5.1 INCOMPLETE PROMPT

You are defining the preconditions and effects (represented in PDDL format) of an AI agent's actions. Information about the AI agent will be provided ...

Before defining the preconditions for an action, consider the implications of the action within the given domain.

Here are two examples from the classical BlocksWorld domain for demonstrating the output format.

<Examples from the original prompt>

Here is the task.
Domain information: {domain_desc}
Action:

Figure 2: An example output of the optimizer LLM that outputs a prompt template instead of a complete prompt. While it is technically correct and successfully added the additional instruction shown in blue, this output is not acceptable since it includes a template holder for examples, which is marked in red, and this output still needs post-processing with the original prompt to make the prompt complete. For simplicity, we have omitted part of the original prompts that is not changed, marked in green, and this part of the prompt can be found in the original paper (Guan et al., 2023).

There are many times that the prompt optimizer fails to generate the complete prompt. As shown in the complete prompt in the appendix, we have already added extensive instructions on generating a complete prompt rather than an incomplete one. As shown in Fig. 2, LLMs sometimes will output a prompt template where it requires a copy-paste done by the users to make the prompt complete, both in our algorithm REPROMPT and our baseline PromptAgent. We found this happens often when the initial prompt is relatively long, and we believe the LLMs are trained to generate a shorter response if possible, but they failed to follow our instructions on generating a complete prompt. To solve this problem, we add an additional LLM to help fill in the template. We also provide the prompt of this LLM in the appendix. This additional LLM helps us to generate a complete prompt in the TravelPlanner domain. We do not use a rule-based fixer because the generated prompt template could be included in multiple symbols including but not limited to <> or {}. To relieve our workload, we choose to let LLM automatically recognize and replace them.

## 5.2 INCORRECT CHANGE BY ACCIDENT

In some domains, the output format can be similar to a more commonly used domain, and LLMs are misled to correct the prompt in certain parts. For example, in our PDDL domains, we ask the LLM to generate the preconditions of the actions rather than the actual PDDL file. In our experiment, we found that even though our prompt has explicitly required the LLM not to change any other part, especially the output format part, the updated prompt still sometimes changes the output format by mistake, specifically, changing the output of "Preconditions" in capital into "precondition" in smaller cases. To solve this problem, we leverage the feedback of the syntax checker. While the generated results could have some errors, the results should always be complete, and have the

required syntax. And if our syntax parser that extracts the answer from the LLM output cannot find the word "Precondition", we know the prompt used is not correct, then we re-run REPROMPT on the same step to generate a correct one. Because the fail rate with our current code is empirically less than 10%, this ad-hoc solution is enough.

## 6 CONCLUSION

In this paper, we have focused on optimizing the prompts used in LLM agents. We propose a new automatic prompt optimizer, REPROMPT, which is based on the summarization of the interaction between LLM agents and feedback providers. Our experiments show that the LLM agents could benefit from an updated prompt in both 1-epoch settings and 5-epochs settings. We also discuss the current limitations of our paper and how future work can adjust them.

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

## A  LIMITATIONS

It is important to discuss the limitations of our proposed method REPROMPT. First of all, as the prompt is optimized similarly to fine-tuning, our generated prompts are also limited to the training data and harm the generalizability of LLMs to a certain degree, i.e., if the training data demonstrate unique challenges to LLMs that only occur in those training scenarios but not general cases, our optimized prompts may even be less efficient compared to the original prompts.

Next, our prompts rely on comprehensive tools available to the LLM agents. Because the optimized methods are provided directly from LLMs rather than processed with a search-based method, REPROMPT can propose to use some statistic tools that are not available in the actual given settings. We leave the possibility of letting LLM code the extra, not-available, but commonly used tools themselves to future works.

Furthermore, sometimes, the feedback generator, which we do not have and put any control on, can generate useless or even wrong and misleading results. While REPROMPT is based on summarization, REPROMPT will take such feedback into the prompt if such a mistake is often made. And because we do not consider removing useless steps in the prompt in this paper, such a mistake will only increase the total number of tokens used, not contributing to better results. Future work could add a search-based mechanism to identify such a mistake and potentially fix it, but this will potentially require more ground-truth feedback from the environment and can lead to a more constraint in applicable domains.

Last but not least, our proposed method is doing the planning in the prompt phase, and thus, if the LLM agents are proposed for very general domains that need completely different procedures in different scenarios, e.g., LLM agents for solving math problems, our proposed method will not work at all. However, if the LLM agents are proposed for specific tasks, like using LLM agents to solve high-school geometry problems, our proposed method could help learn the planning very efficiently, as shown above in the experiments.

## B    IMPLEMENTATION DETAILS FOR PROMPTAGENT BASELINE

PromptAgent Wang et al. (2024) was initially proposed as an APE algorithm for question-answering tasks, which made its design mostly use the fact of ground-truth feedback provider which tells at least whether the final prediction is correct or not. In our TravelPlanner settings, even this boolean checker is not provided by the environment during the process of APE.

To adapt the PromptAgent framework to our settings, we created an LLM-based checker, which provided exactly the same input as the plan generator together with the plan just generated. The checker is called after the reflexion finished and submitted a plan, i.e., each node in the MCTS tree of PromptAgent is now a complete reflexion run. We provide the prompt of the checker in Fig. 3. During testing, we compared the performance of the LLM-based checker with a ground-truth checker, which is only available under specific dataset test conditions. The accuracy of the LLM-based checker is 89%. Although the 89% accuracy might seem reasonable, it's important to note that the success rate of a plan is less than 5%. This means that even a checker that consistently marks plans as incorrect would achieve better performance overall. We observe the same type of hallucinations as the ones in the plan generation, which is one of the main reasons that there are many false negatives in this checker. Meanwhile, we have briefly tested 5 of the false negatives with the latest GPT-4o, and we found that, surprisingly, they are all correct. However, to make a fair comparison and for the consistent of models used in the experiment, we are still using GPT-4-turbo as the model for the checker.

Additionally, we have made some necessary changes to the gradient descent prompt template to remove the requirement of labels, also known as the ground-truth answers, which in our case is also not provided.

To make the comparison between REPROMPT and PromptAgent fair, we use the Lite version of PromptAgent to limit the number of iterations. However, even with PromptAgent-Lite, it is still about twice as expensive compared to REPROMPT, which shows another advantage of our algorithm.

## C    PSEUDO CODE FOR REPROMPT

While in the main paper, we only provided the workflow of our paper, here we provide actual pseudo code. While REPROMPT can be an analogy to fine-tuning on the prompt space, the code is very similar to a typical ML training loop as shown in Alg. 1.

---

**Algorithm 1** REPROMPT train loop

---

1: **function** TRAIN
2:     $Prompt \leftarrow InitialPrompt.$
3:     **for** batch, x from Dataloader **do**
4:         $Response \leftarrow Act(Prompt, x)$
5:         $Loss \leftarrow Summarize(Response, x, Prompt)$
6:         $Prompt \leftarrow Optimizer(Loss, Prompt)$
7:     **end for**
8:     **return** Prompt
9: **end function**

---

You are an AI assistant. Your job is to determine whether a specific travel plan meets the constraints. The constraints are provided below. You will be provided with some information about the candidate trip contents, a specific query, and a proposed solution to the query. Please analyze the query and evaluate whether the query is correct or not. Note that all the information in the plan should be derived from the provided data, and do not do any additional estimation or approximation. Put your final judgment of "Correct" or "Wrong" in a \bbox{}.

Constraint Description

Environment Constraint

Unavailable Transportation: There is no available flight or driving information between the two cities.
Unavailable Attractions: There is no available attraction information in the queried city.

Commonsense Constraint

Within Sandbox: All information in the plan must be within the closed sandbox; otherwise, it will be considered a hallucination.
Complete Information: No key information should be left out of the plan, such as the lack of accommodation during travel.
Within Current City: All scheduled activities for the day must be located within that day's city(s).
Reasonable City Route: Changes in cities during the trip must be reasonable.
Diverse Restaurants: Restaurant choices should not be repeated throughout the trip.
Diverse Attractions: Attraction choices should not be repeated throughout the trip.
Non-conf. Transportation: Transportation choices within the trip must be reasonable. For example, having both "self-driving" and "flight" would be considered a conflict.
Minimum Nights Stay: The number of consecutive days spent in a specific accommodation during the trip must meet the corresponding required minimum number of nights' stay.

Hard Constraint

Budget: The total budget of the trip.
Room Rule: Room rules include "No parties," "No smoking," "No children under 10," "No pets," and "No visitors."
Room Type: Room types include "Entire Room," "Private Room," "Shared Room," and "No Shared Room."
Cuisine: Cuisines include "Chinese," "American," "Italian," "Mexican," "Indian," "Mediterranean," and "French."
Transportation: Transportation options include "No flight" and "No self-driving."
==== Given information: {information}
Query: {query}
The generated plan is: {plan}

Figure 3: The checker prompt for PromptAgent. The prompt should be fed into an LLM, which in this paper is GPT4-turbo, to get the judgement of whether the current result is correct or wrong.

**System Prompt**

You are a summarizer. You wil be provided with a chat history from an AI assistant and the user. Please choose one of the following that you believe is the case, and summarize the focus point as instructed:

a). You can summarize the main reason for failures that led to this length of discussion. You only need to summarize the reason that has appeared, but not further summarize and infer the reason behind all the reasons. Make sure you choose only one reason at a time.

b). There is a specific thought or a list of similar thoughts that is very helpful to getting the correct answer. In this case, try to generalize the thought and make it does not involve detail information like concrete numbers, but as a high-level thought of what aspect should be highlighted and focus on.

c). There is no general reason that leads to a failure. It is case-by-case errors that is inevitable. First, do some short analysis, and then finish your conclusion in one single line, starting with: "In conclusion, the main focus point should be: "

**User Prompt**

Here is the chat history, please follow the instructions above and tell me what is the main focus point should be in the required format:

<Chat History>

Figure 4: The loss summarize prompt. The prompt should be fed into an LLM, which in this paper is GPT4-turbo, to get the loss used to optimize the prompt.

You are a template replacer. You will be provided with an original prompt, and an optimized prompt. Part of the new optimized prompt is a placeholder that needs to be replaced with the original prompt. Your job is to replace the placeholder with the original prompt.

One example of the placeholder is: " ⟨ Original Prompt Start ⟩ ". You need to replace this placeholder with the original prompt.

Another exmpale is <Examples from the original prompt>. You need to replace this placeholder with the examples from the original prompt.

Output directly the new prompt with the placeholder replaced. Do not provide any additional note or analysis.

Figure 5: The prompt to fix the place holders in the optimized prompt.

> **System Prompt**
>
> You are a prompt optimizer. You will be provided with an original prompt, and a specific point that this round of optimization should focus on. Your job is to update the prompt based on the provided focus point. If the focus point is saying there is no general reason, then skip all the following step and directly output the original prompt.
>
> In the process, do the following steps one by one:
>
> 1. List a few different options that could address the given focus point.
> 2. Choose the solution that you think is the most promising. Make sure the solution is focus on instruction on how to solve the problem rather than instructions on giving better problem description. The solution should not be too general and should bring in actual insights.
> 3. Analyze each steps in the original prompt, and see whether the new solution should be inserted before or after the current step, or it is a superset of the current step and thus the original step should be replaced.
> 4. Finish your output with your final prompt, in the format of: "Based on the above analysis, the improved prompt is: ".
>
> A few common solutions for specific problems are:
>
> - If some details are missed, a sentence by sentence check ahead of time could be helpful.
> - If some requirement are not meet, then a first analysis on that constraint could be helpful, or keep satisfying that requirement in mind when giving the solution could be useful.
> - If it is already a thought, then a check on whether the thought is still workable in the given scenario is very helpful. For example, if it is about a speicific requirement need to be meet, then maybe also make sure to check it in every step. However, make sure this does not limit what the feedback can provide, and using words like "specifically" to remind such a check.
>
> During the process, make sure that you focus on optimizing the prompt for the given focus point, and do not provide any additional information.
>
> Do not change any other part of the prompt. Only focus on the step-by-step instructions. Especially, do not change the examples and the format requirement. However, make sure you copy the detailed previous example completely to the new output instead of using place holders to indicate that it should not be changed. Do not worry about the output length caused by the examples.
>
> Please provide a detailed and complete response without omitting any information or use "..." or "[...]"to replace any part of the prompt. Again, ensure that no information is omitted or summarized.

Figure 6: The prompt optimizer prompt. The prompt should be fed into an LLM to update the prompt for problem solving.

> You are defining the preconditions and effects (represented in PDDL format) of an AI agent's actions. Information about the AI agent will be provided in the domain description. Note that individual conditions in preconditions and effects should be listed separately. For example, "object_1 is washed and heated" should be considered as two separate conditions "object_1 is washed" and "object_1 is heated". Also, in PDDL, two predicates cannot have the same name even if they have different parameters. Each predicate in PDDL must have a unique name, and its parameters must be explicitly defined in the predicate definition. It is recommended to define predicate names in an intuitive and readable way.
> Here are two examples from the classical BlocksWorld domain for demonstrating the output format.
> <Examples from the original prompt>
> Before defining the preconditions for an action, consider the implications of the action within the given domain. Identify any additional preconditions that are critical for the action to be performed successfully. Ensure that all necessary conditions are accounted for before listing them.
> Here is the task.
> Domain information: {domain_desc}
> Action:

Figure 7: The optimized prompt for PDDL generation. The main changes are highlighted in blue.

## D PROMPTS FOR REPROMPT

Here, we provide all the prompts used in our paper. The prompt used to summarize the loss is provided in Fig. 4, the prompt used to optimize the prompt is provided in Fig. 6, and the prompt used to replace the placeholders that could be accidently included, which is discussed in Sec. 5, is provided in Fig. 5.

## E OPTIMIZED PROMPT FROM REPROMPT

Here, to help reproducibility, we provide all the optimized prompts that leads to the results shown in Table. 1 and Table. 2. It needs to be addressed that even if the prompt, the temperature, the model used, and the seed are the same, OpenAI APIs still do not guarantee that the generated output will be exactly the same every time. This will greatly affect the final results in our case, given that REFLEXION (Shinn et al., 2023) is used to provide the feedback, and a small change on the earlier reflection can lead to completely different results in the end. Due to unknown reasons, we found that the reflection module in REFLEXION (Shinn et al., 2023) can provide completely useless and even wrong suggestions to the LLM and lead to very bad results. If this happens, we suggest rerunning the baseline model without RePrompt to make sure the OpenAI is providing correct feedbacks. However, the final conclusion, especially the relatively superior of our model, is always found to be true in our experiments. And for a fair comparison and to match the results from the TravelPlanner paper, all the results reported in the paper are the best among the 3 trials over time (Best-of-3).

In Fig. 7, we provide the optimized prompt for PDDL action generation. In Fig. 8 and Fig. 9, we provide the optimized prompt for the TravelPlanner environment (Xie et al., 2024). In Fig. 10, we provide the optimized prompt generated by PromptAgent Wang et al. (2024) for TravelPlanner environment.

You are a proficient planner. Based on the provided information and query, please give me a detailed plan, including specifics such as flight numbers (e.g., F0123456), restaurant names, and hotel names. Note that all the information in your plan should be derived from the provided data. You must adhere to the format given in the example. Additionally, all details should align with common sense. Attraction visits and meals are expected to be diverse. The symbol '-' indicates that information is unnecessary. For example, in the provided sample, you do not need to plan after returning to the departure city. When you travel to two cities in one day, you should note it in the 'Current City' section as in the example (i.e., from A to B). Before starting the planning process, establish a budget breakdown for each category (transportation, meals, attractions, accommodation) to ensure that the total cost does not exceed the provided budget. Solve this task by alternating between Thought, Action, and Observation steps.
The 'Thought' phase involves reasoning about the current situation and specifically the budget constraints.
The 'Action' phase can be of two types:
(1) CostEnquiry[Sub Plan]: This function calculates the cost of a detailed sub plan, which you need to input the people number and plan in JSON format. The sub plan should encompass a complete one-day plan. An example will be provided for reference.
(2) Finish[Final Plan]: Use this function to indicate the completion of the task. You must submit a final, complete plan as an argument.
***** Example *****
<Examples>
***** Example Ends *****
{reflections}
You must use Finish to indict you have finished the task. And each action only calls one function once.
Given information: {text}
Query: {query}{scratchpad}

Figure 8: The prompt of TravelPlanner optimized after 1 epoch of REPROMPT. The main changes are highlighted in blue.

You are a proficient planner. Based on the provided information and query, please give me a detailed plan, including specifics such as flight numbers (e.g., F0123456), restaurant names, and hotel names. Before you start planning, conduct a preliminary budget analysis to understand the cost constraints for each category (transportation, accommodation, meals, and attractions). Ensure that the accommodation information is formatted according to the predefined template compatible with the cost calculation environment. After setting the preliminary budget, conduct a comparative analysis of transportation options to select the most cost-effective one, research meal options to find the best value that fits dietary preferences and proximity requirements, and compare accommodation choices based on cost, location, amenities, and reviews. Set specific budget limits for meals and accommodations to ensure the overall expenses do not exceed the budget while maintaining a satisfactory experience. Ensure that each choice of transportation, accommodation, meal, and attraction is tailored to the specific preferences and requirements provided in the query, making iterative adjustments to the plan as necessary to stay within budget constraints.

Note that all the information in your plan should be derived from the provided data. You must adhere to the format given in the example. Additionally, all details should align with common sense. Attraction visits and meals are expected to be diverse. The symbol '-' indicates that information is unnecessary. For example, in the provided sample, you do not need to plan after returning to the departure city. When you travel to two cities in one day, you should note it in the 'Current City' section as in the example (i.e., from A to B). Before starting the planning process, establish a budget breakdown for each category (transportation, meals, attractions, accommodation) to ensure that the total cost does not exceed the provided budget. Solve this task by alternating between Thought, Action, and Observation steps.

The 'Thought' phase involves reasoning about the current situation and specifically the budget constraints.

The 'Action' phase can be of two types:
(1) CostEnquiry[Sub Plan]: This function calculates the cost of a detailed sub plan, which you need to input the people number and plan in JSON format. The sub plan should encompass a complete one-day plan. An example will be provided for reference.
(2) Finish[Final Plan]: Use this function to indicate the completion of the task. You must submit a final, complete plan as an argument.
***** Example *****
<Examples>
***** Example Ends *****
{reflections}
You must use Finish to indict you have finished the task. And each action only calls one function once.
Given information: {text}
Query: {query}{scratchpad}

Figure 9: The prompt of TravelPlanner optimized after 5 epochs of REPROMPT. The main changes are highlighted in blue.

You are a proficient planner with a keen eye for detail and practicality. Your task is to create a comprehensive travel plan that adheres to the provided budget and timeframe, ensuring a diverse and enjoyable experience. The plan should include specific flight numbers, restaurant names, hotel names, and attraction details, all of which must be derived from the provided data. Follow the format shown in the example, and ensure that all details are sensible and feasible. When planning, consider the following guidelines:
- Ensure meal diversity by not repeating restaurant choices for different meals.
- Select transportation options that are practical and feasible, considering the distance and geography.
- Provide complete information, including all meals for each day and any in-city transportation if necessary.
- Adhere to the budget, allocating funds across flights, accommodations, meals, and attractions.
- Check for any minimum stay requirements or house rules for accommodations.
- Use the symbol '-' to indicate when information is unnecessary, such as after returning to the departure city or when no transportation is needed within the current city.
Your planning process should consist of alternating Thought, Action, and Observation steps:
- Thought: Reason about the current situation and what needs to be planned next.
- Action: Perform one of two types of actions: (1) CostEnquiry[Sub Plan]: Calculate the cost of a detailed sub-plan for a complete one-day plan. Input the number of people and the plan in JSON format.
(2) Finish[Final Plan]: Indicate the completion of the task by submitting a final, complete plan as an argument.
- Observation: Reflect on the information received from the actions and adjust the plan accordingly.
Remember, each action should only call one function once, and you must use Finish to indicate you have finished the task.
Here is an example for reference:
***** Example *****
{Example content}
***** Example Ends *****
Now, let's begin planning based on the given information and query. Keep in mind that the plan should be logical, feasible, and within the specified constraints. Good luck!
{reflections}
Given information: {text} Query: {query}{scratchpad}

Figure 10: The prompt of TravelPlanner optimized by PromptAgent. Unlike REPROMPT, majority prompt has been changed and thus we do not do further highlight.

