# OpenReview forum: "RePrompt: Prompt Engineering for Large Language Models Agents through Reflection"
_ICLR.cc/2025/Conference — Submitted to ICLR 2025_

### Official Review · Reviewer_vPhR · 2024-11-01

**Soundness:** 2
**Presentation:** 1
**Contribution:** 1
**Rating:** 3
**Confidence:** 3

**Summary:**

This paper proposed a new method called RePrompt, a prompt optimization procedure specially tailored for planning domains. RePrompt combines Act procedure (a method used in previous literature to improve the reasoning by providing reflection step-by-step) and prompt optimization.

RePrompt is evaluated on two task planning domains: PDDL Generation and Travel Planning. In PDDL Generation domain, RePrompt demonstrate reduced incorrect actions and errors compared to the original paper. In the Travel Planning domain, RePrompt is compared with Reflexion and PromptAgent with better success rates across all groups.

**Strengths:**

The performance improvement over the baselines are rather strong.

**Weaknesses:**

1. The method can be understood as an combination of ReAct and Automatic Prompt Optimization. However, it is not clear why the Act part that provides feedbacks during generation of the plan can improve the prompt optimization. Further ablation study is necessary, for example, optimizing prompts without Act and inferencing with/without Act in figure 1.

2. The paper does not have any formulation or an algorithm box. Some descriptions are very vague and the references to some components are not consistent. For example, in figure 1, the work flow involves a LLM summarizer, but line 186 indicates using a summarization team. Also, line 182 mentions "focus point". This is very illy-defined. If it is emerged from a prompt, the author should explicitly show the prompt that instruct LLMs to label "focus point". More vague descriptions like "some interaction schemes with some kind of feedback provider" (line 179). Such process should be more precisely described with formulation and algorithm box. Right now, I don't believe the Methods Section is describing a method that is replicable and able to be applied to other domains by the community.

3. The difference with existing methods are not clear. For example, the reference to ProTeGi says "Compared to previous work in automatic prompt engineering, including APE (Zhou et al., 2023) and PROTEGI (Pryzant et al., 2023), we do a summarization over a batch, and we prevent the prompt optimization from overfitting on a single outlier data point." This reference is incorrect as ProTeGi also optimizes the prompt over a minibatch.

**Questions:**

1. I would appreciate a more clear presentation of the Method section. The author should clearly indicate the LLMs and human summarization team. Different LLMs should be referred to with consistent names with the prompt used prompt them acting their roles in the pipeline

2. The framework is designed for optimizing prompt for LLM agents, for example, those involve Act Procedure. Could author explain how the method is accommodated to LLM agent that involves multiple rounds generation and more than one LLMs? After I reading the method section, the Act part can be removed and the rest of the system is exactly the same as previously prompt optimization framework.

---

> ### Author Response · Authors · 2024-11-16
> **Rebuttal**
>
> Thank you for your review. Here we address both your concerns in the weakness (W1-W3), and the ones in questions (Q1, Q2), which we found is highly correlated to the weakness.
>
>
> W1. Q2. We combine these two points together as they both refer to how the Act framework, i.e., the setting of LLM Agents, relates to our paper. There are two main differences caused by such settings. First of all, LLM agents are normally used in complex tasks where the final checker might not be available or might be very pricey. This is why much previous work in APE cannot be used in these tasks—because of the cost. The second difference is that even if there is a final checker in the Act framework, it could be much less informative than intermediate feedback, and thus the need for intermediate feedback is very important, which is why our method focuses on summarizing the intermediate feedback as guidance. There is also a minor point that is not very general to the whole domain but affected our design choice: some tasks in this domain require a strict format to interact in the Act framework. If we allow the prompt optimizer to change randomly, it will waste a lot of time changing to incorrect formats. To resolve this, our method focuses on changing one step in the CoT steps.
>
>
> W2. Q1. These two points concern the clarity of the methods section. We have revised the method description in the updated version of the paper. Due to the page limit, we were unable to include all the prompts, which are highly relevant to the details in the methodology section, in the main paper. However, we have updated the paper to make this section easier to read and more self-contained. We invite you to review the updated version to verify these changes. Here, we provide direct answers to the specific points you raised:
> - We found the use of the term “summarization team” to be confusing, as the summarizer is still an LLM. We have changed it to “summarizer.”
> - The “focus point” is indeed a crucial detail in the optimizer prompt. We have included the prompt in the appendix and have highlighted the relevant part in the updated version of the paper.
> - Regarding the feedback provider, we deliberately kept the description high-level, as our approach does not impose any specific constraints on this component. However, we acknowledge that this led to an undesirably vague description, rather than helping readers understand the generalizability of our method. We have revised the wording in the section you mentioned and clarified this point in a separate paragraph in the methodology section later in the paper.
> - Regarding the algorithm box, we have included pseudocode in our appendix. While our method uses LLMs and the prompts themselves cannot be included in the code due to their length, we do not see a significant difference between the pseudocode and the workflow figure already included in the paper. As this is a paper on large language models, we believe it is unnecessary to introduce notations and formulations that do not improve readability. Therefore, in this updated version, we still have not included any formulations.
>
>
>
>
> W3. Thank you for pointing this out. We found this sentence very misleading as our original point to make is to highlight the summarization step rather than the batch. We have updated our paper accordingly to avoid this misunderstanding.

---

### Official Review · Reviewer_TG95 · 2024-11-01

**Soundness:** 2
**Presentation:** 2
**Contribution:** 2
**Rating:** 5
**Confidence:** 4

**Summary:**

This paper proposes an approach for automatic prompt engineering for cases where end evaluation may be expensive or infeasible, but it may be easier to get intermediate feedback on the generated prompts. The proposed approach works in a step-by-step manner which takes the chat history 'interactions' and `reflections' along with an internal LLM evaluation into consideration for improving the prompt at every step. The authors show experimental results on the PDDL file generation domain and the travel planning domain showing improvements on reduced total number of errors and increased success rate across the two domains, respectively.

**Strengths:**

1) The paper aims to approach an interesting problem of optimizing prompts for reasoning tasks where external evaluations are expensive/infeasible to get, making the prompt optimization step harder.
2) The proposed approach is well-explained with the help of an example in Section 1, which allows the readers to get a good understanding of the prompt optimization flow.
3) Section 5 consists of a detailed error analysis, which unfortunately is not seen in many LLM papers, and is therefore a very refreshing and useful component of the work in its current stage.

**Weaknesses:**

1) Novelty: One of the biggest concerns in the current version of the work are in the novelty/contribution of the claims. The authors state (specifically in line 102, along with other places in abstract and Introduction) that the proposed approach uses a `summarization-based' to improve the prompt at any given step. However, please note that this is indeed what the core idea of Reflexion [1] is, which suffers from its own drawbacks due to the dependence on ReAct for exemplar-query similarity as shown in [2]. It is hard to understand from the current text what the difference is, other than this work being an application of Reflexion for APE. It would be helpful if the authors can clearly highlight/elaborate on this claim, or present this work as being Reflexion's application for APE (which still seems to be novel, but needs to be said upfront if that is the case.)

2) Abuse of terminology: The authors use terms such as as 'gradient-based' and 'standard fine-tuning procedure in ML' and 'loss optimization', however, the approach does not seem to be fundamentally the same as these. If the authors intend to use these terms as an analogy for easier understanding of the readers, that needs to be made clear in the paper. Although, it still seems to be a very far-fetched and overused analogy throughout the paper. If the authors believe that the reviewer misunderstood the usage of these terms for the proposed approach, it will be really helpful to have a transparent explanation of how these terms make sense in the given context of APE, what is the gradient-based approach, what is the mathematical loss function used, etc.

3) Both PDDL generation and Travel Planning have sound verifiers/checkers, which is contrary to the author's claims and problem setting for the proposed approach. It will be helpful if the authors can provide evidence on why these verifiers are unavailable (due to a modified problem setting perhaps). In the current stage of the work, it does not seem that the paper's motivation is coherent with the experimental settings used. Additionally, the work done by [3] has far superior results than the baselines used in the work, and it would be useful to know why the authors did not use it as a baseline or perhaps not consider it as a reasonable baseline for this work.


[1] Shinn, Noah, et al. "Reflexion: Language agents with verbal reinforcement learning." Advances in Neural Information Processing Systems 36 (2024).
[2] Verma, Mudit, Siddhant Bhambri, and Subbarao Kambhampati. "On the Brittle Foundations of ReAct Prompting for Agentic Large Language Models." arXiv preprint arXiv:2405.13966 (2024).
[3] Gundawar, Atharva, et al. "Robust Planning with LLM-Modulo Framework: Case Study in Travel Planning." arXiv preprint arXiv:2405.20625 (2024).

**Questions:**

1) The entire motivation of the work rests on the fact that there are cases where final prompt evaluations may be expensive or infeasible. Could the authors provide examples of such problem settings and domains, and clearly explain why the respective evaluations are not easily available?
2) Line 312: what is meant by `train RePrompt'? What exactly is the training step and what is being learnt in this case?
3) Why are there no other APE methods used as baselines in the experimental setting?

---

> ### Author Response · Authors · 2024-11-16
> **Rebuttal (1/2)**
>
> Thank you for your review. Here we respond to the weaknesses you listed one by one:
>
>
> W1. We agree that our method, REPROMPT, incorporates knowledge from the self-reflection phase into the prompt engineering phase. This approach helps to avoid early-stage errors and, more importantly, reduces the randomness observed in methods like ReAct and Reflexion, where specific scenarios may not receive adequate feedback. By summarizing common issues, such as budget balancing, REPROMPT ensures these are addressed in the prompt, allowing ReAct and Reflexion to focus on case-specific issues, like unexpected price spikes for hotels in certain cities on particular days. Since this was a common question among reviewers, we have included this discussion in our revised paper, which we invite you to review for further clarification. Specifically, our approach operates at a level above Reflexion by summarizing feedback generated by Reflexion and optimizing the prompt accordingly. This explains why REPROMPT outperforms in cases like TravelPlanner, where the feedback generator is Reflexion—because Reflexion summarizes ReAct, and REPROMPT, in turn, summarizes Reflexion. Moreover, we highlight that our paper is the first to address prompt engineering in settings where the final checker is unavailable during optimization. This, as noted in your Q1, is a novel contribution. We consider our main contribution to be the study of this unique setting, with the adaptation of Reflexion to APE being a natural development within this context.
>
>
>
>
> W2.  Thank you for your suggestion. We have changed correspondingly to help the readers understand the analogy, and the detailed analogy in Sec. 3.2. Again, we invite you to take a look at the corresponding paragraph, especially the second paragraph of section 3.2, to verify our change. And we have also provided a copy of that paragraph to the Q2 below for your reference.
>
>
> W3. The feedback in the PDDL benchmark is provided by human experts, which is very time-consuming. The average judging speed is approximately 20 responses per hour per expert, making it too slow to be used as an intermediate process for APE. For TravelPlanner, while the checker is available, the dataset authors have explicitly stated that such a symbolic method should not be used in the process (https://github.com/OSU-NLP-Group/TravelPlanner?tab=readme-ov-file#%EF%B8%8Fwarnings). This is also why the other work [3] achieves much superior results, as they use external code executors that encode the constraints to facilitate the generation process, which completely undermines the original purpose of the dataset. To adhere to the requirements and purpose of the datasets and avoid exploiting their limitations, we do not compare that paper as a baseline.
>
>
> Q1. There are numerous cases where an accurate final evaluator is highly costly, making it impractical for most people to use them frequently during training or, in our case, while optimizing the prompt. This is especially common in high-specialization domains that demand extensive domain knowledge. For example, PDDL generation, included as a dataset in our paper, requires such expertise. Similar challenges arise with LLM applications in fields like physics, chemistry, and other scientific disciplines. This need for high-quality data generation has led to the emergence of startups like scale.ai, which support training LLMs by providing reliable datasets. There are also scenarios where a final evaluation might be entirely absent, as seen in applications like ChatGPT, and particularly in GPT's tools. In these situations, users interact with the application, sometimes providing intermediate feedback, but often leaving without clarifying whether they received a satisfactory response or abandoning the interaction due to dissatisfaction with the model’s ability to assist further. Our method allows such applications to optimize their prompts even in the absence of explicit final evaluations, which was previously unachievable.

---

> > ### Author Response · Authors · 2024-11-16
> > **Rebuttal (2/2)**
> >
> > Q2. We believe that this is a question that is highly correlated with the weakness 2. So, we again invite you to look into our revised version of the paper. Here, we have attached the corresponding paragraph that seems to have directly answered your question.
> >
> >
> > > As shown in Fig. 1, our method, REPROMPT, is a prompt optimizer that is based on the interaction-based action generate process. Our method is similar to a machine learning training loop, which iterates between getting the output based on the current parameter, calculating the loss based on the output, and optimizing the parameter based on the loss. But in our case, the parameters to be trained are the prompts going to be fed into the model, the model forward pass is replaced by the complete interaction-based action generate process, which includes the feedback information generator, the loss and optimizer are both LLMs instead of numerical calculation for the distance and the gradients.
> >
> >
> > Q3. As we stated in W3, in PDDL, the checker is very costly and time-consuming, making it impractical for methods that require many iterations. In TravelPlanner, we have already compared our method to one of the most commonly used and, in most cases, the strongest baseline, PromptAgent [R1], to demonstrate the superiority of our approach.
> >
> >
> > [3]. Gundawar, Atharva, et al. "Robust Planning with LLM-Modulo Framework: Case Study in Travel Planning." arXiv preprint arXiv:2405.20625 (2024).
> >
> >
> > [R1]. Xinyuan Wang, et al. Promptagent: Strategic planning with language models enables expert-level prompt optimization. In The Twelfth International Conference on Learning
> > Representations, ICLR 2024, Vienna, Austria, May 7-11, 2024.

---

> > ### Comment · Reviewer_TG95 · 2024-11-18
> > **Response**
> >
> > Thank you for your answers to the mentioned weaknesses and questions!
> >
> > W1. The example provided to distinguish the extra step that RePrompt does from Reflexion is somewhat helpful, but the contribution does not seem to be novel enough. Reflexion (as a follow-up from ReAct) still suffers from the limitation of providing examples in the prompt which are egregiously similar to the problem query (Please see my initial comment for the references). As an extension, RePrompt would face a similar issue as well to the best of my understanding. While I agree that RePrompt allows for searching for a better prompt over multiple iterations, this example dependency makes the initial step extremely demanding and costly for the prompt engineer.
> >
> > W2. The updated write-up in Section 3.2 is very helpful and much easier to understand the analogy now, thanks.
> >
> > W3 and Q3. I can agree with the human-intensive process for validating the PDDL benchmark solutions, but disagree with the availability of final evaluations available in the TravelPlanner benchmark. Even in [3], the authors seem to have used evaluators only to provide feedback, which is information available from the dataset. Also, giving LLMs the complete benefit of doubt and providing them will all the information upfront, my understanding is that they would still have to extract the problem-specific information and then reason over that to generate the final travel plan. Moreover, it is much more reasonable to assume that feedback is hard to obtain during the intermediate steps but only available at the end - a property that is easily seen in several simulation and real-world settings. Ideally, it would have been helpful to have a domain in the work which is more representative of this setting.

---

> > > ### Author Response · Authors · 2024-11-18
> > > **Further Responses**
> > >
> > > Thank you for your fast response. Here, we address your remaining concerns in W1 and W3 (which we have divided into two parts).
> > >
> > >
> > > W1. We agree that our method will face similar challenges on the reflection side, as we do not impose any constraints or modifications on how the reflection is generated. However, we want to emphasize that, as demonstrated in our paper, our method is not limited to ReAct/Reflexion and can leverage any framework capable of providing feedback. This means that any future advancements in this direction could be integrated into the RePrompt framework to address these challenges. More importantly, our work is entirely novel within the automatic prompt engineering community, being the first to eliminate the need for a ground-truth checker in the process. We want to clarify that the objective of comparing RePrompt and Reflexion in the paper is not to assert that we are a much superior reflection algorithm but to demonstrate that prompt engineering can enhance the reflection framework, and that our RePrompt is a novel algorithm capable of achieving this automatically.
> > >
> > >
> > > W3. We want to further highlight that in [3], the performance results are based on outcomes after multiple iterations with the critics (used to provide feedback, as you mentioned). This approach is strictly prohibited since the symbolic evaluators should be used only during testing, meaning it is not allowed to serve as a feedback provider, as stated in the link we provided earlier. In contrast, our work strictly adheres to the dataset requirements, using Reflexion as the feedback provider instead of the evaluator. The original questions are straightforward and formatted in human language, such as: "Please create a travel plan for me where I'll be departing from Washington and heading to Myrtle Beach for a 3-day trip from March 13th to March 15th, 2022. Can you help me keep this journey within a budget of $1,400?" (as shown in Fig. 1 of our paper). While the original prompt is high-level and only mentions that the answer should follow common sense, using all the information upfront has significantly helped the LLM identify the exact problems in the provided solutions and extract hidden constraints in the problem. This fundamentally changes the nature of the TravelPlanner dataset, turning it into a completely different and much easier problem to solve. We are happy to add this work to our related work section and clarify the differences between our settings, as discussed above. However, we do not believe the results are directly comparable.
> > >
> > >
> > > >. Moreover, it is much more reasonable to assume that feedback is hard to obtain during the intermediate steps but only available at the end.
> > >
> > >
> > > We want to clarify that, in this paper, we are not considering feedback as being available at the end but rather in a scenario where it is only available during test time. We provided a list of examples in the Q1 section of our previous response, but unfortunately, we have not yet received your feedback on them. Across all the examples we provided, our RePrompt method can optimize the prompt, whereas previous automatic prompt engineering approaches cannot be applied. In the paper, we selected the current two datasets from among all the possible options because of their unique properties regarding feedback quality and availability.
> > >
> > >
> > > We hope this helps address your concerns. Please let us know if you have any further questions.

---

> > > > ### Comment · Reviewer_TG95 · 2024-11-19
> > > > **Response**
> > > >
> > > > Thanks again for your clarifications!
> > > >
> > > > W1. I understand that this is not a direct comparison of RePrompt with Reflexion, but given the current setup in the paper, there is no way to assess how the method works if not utilized on top of React/Reflexion. It would be unfair to assess the paper based on 'how potentially it can be used' with any framework capable of providing feedback. Another baseline feedback approach can bring its own issues to the table, but what would be constructively useful is how RePrompt can further eliminate those. In the current scope of the work, it could be helpful to see how RePrompt can alleviate some of the underlying issues of Reflexion too. I completely understand the novelty part of the work in the APE community but once again, there don't seem to be any baselines, as mentioned in my initial review. While they may be using a different setup, it is still worthwhile to show any comparisons than none at all, which could have potentially given more credit to the proposed approach.
> > > >
> > > > W3. I disagree that the access to verifiers changes the nature since the reasoning problem still persists that requires the LLM to solve it. Despite the disagreement, the authors don't have to necessarily cite the mentioned work if they believe it is not relevant. It can in fact be useful to include other prompt engineering works that have possibly tested on the TravelPlanner benchmark.
> > > >
> > > > Q1. My apologies for missing out the response earlier. I am convinced that the motivation for datasets that do not have final evaluations available is very reasonable and the authors' responses are helpful to understand the use of the given datasets.
> > > >
> > > > I have updated my score based on the rebuttal discussion, thanks!

---

> ### Author Response · Authors · 2024-11-19
> **Further Response**
>
> Thank you again for your prompt response. We are pleased to see that you agree with our previous discussion. Here again we address your remaining concerns in W1 and W3.
>
>
> W1. We apologize if our previous response may have confused you by conflating the baselines and the dataset settings (and perhaps also due to the fact that we needed to split the initial rebuttals into two parts because of character limits). However, we would like to kindly refer you to our previous response to Q3, where we clarified why there are no baselines in the PDDL generation benchmark. In TravelPlanner, we have already included PromptAgent as our baseline, which seems to contradict your claim that “there don’t seem to be any baselines.” If you are referring to other baselines outside the APE domain, please let us know, and we would be happy to include them if they could significantly improve our paper.
>
>
> W3. While the baseline discussion is included in our response to W1 above, here we further highlight the difficulty caused by the lack of access to verifiers and other informations. To illustrate this, we would like to first quote the example question again, along with the initial prompt of TravelPlanner provided by the dataset creator:
>
>
> > Please create a travel plan for me where I'll be departing from Washington and heading to Myrtle Beach for a 3-day trip from March 13th to March 15th, 2022. Can you help me keep this journey within a budget of $1,400?
>
>
> > You are a proficient planner . Based on the provided information and query , please give me
> a detailed plan , including specifics such as flight numbers (e.g., F0123456 ) , restaurant
> names , and hotel names . Note that all the information in your plan should be derived from
> the provided data . You must adhere to the format given in the example . Additionally , all
> details should align with common sense . Attraction visits and meals are expected to be
> diverse . The symbol '-' indicates that information is unnecessary . For example , in the
> provided sample , you do not need to plan after returning to the departure city . When you
> travel to two cities in one day , you should note it in the ' Current City ' section as in
> the example ( i.e., from A to B). Solve this task by alternating between Thought , Action ,
> and Observation steps . The ' Thought ' phase involves reasoning about the current situation .
> The ' Action ' phase can be of two types :
> (1) CostEnquiry [ Sub Plan ]: This function calculates the cost of a detailed sub plan , which
> you need to input the people number and plan in JSON format . The sub plan should
> encompass a complete one - day plan . An example will be provided for reference .
> (2) Finish [ Final Plan ]: Use this function to indicate the completion of the task .
> You must submit a final , complete plan as an argument .
>
>
> As one can see in these two parts, while some common-sense constraints like “Within Sandbox,” “Complete Information,” and “Diverse Restaurants/Attractions” are explicitly included in the prompt, other important common-sense constraints such as “Non-conflicting Transportation,” “Within Current City,” and “Minimum Nights Stay” are never addressed in the prompt. Providing the information and evaluator ahead of time exposes these hidden requirements to the LLMs, and giving LLMs hints about these hidden common-sense constraints should also be considered within the prompt space. These constraints are so hidden that they are still not included in the optimized prompt, even after applying our RePrompt method or the baseline PromptAgent. We believe this is part of the reason why, in [3], the common-sense constraints macro pass rate is as high as 40.6%, while our method only achieved 6.11%. In comparison, in [3], the authors only reached a 39.4% pass rate for hard constraints, whereas Reflexion alone achieved a 33.89% pass rate. This small gap is likely due to receiving more accurate feedback during the process, which helps eliminate hallucinations (one of the key errors identified in both our Section 5 and the original TravelPlanner paper) and other errors. However, knowing what is being tested and what needs to be considered during reasoning significantly contributes to the greatly improved result on the common-sense pass rate and, consequently, the final pass rate.
>
> Again, we hope this helps address your concerns. Please let us know if you have any further questions.

---

### Official Review · Reviewer_vpmw · 2024-11-04

**Soundness:** 2
**Presentation:** 3
**Contribution:** 2
**Rating:** 3
**Confidence:** 4

**Summary:**

The paper presents REPROMPT - where the authors focus on leveraging intermediate feedback (which they claim to be more readily available then final evaluator). The idea is to obtain some potential solution (LLM generated) of a given problem with intermediate feedback. These (solution, feedback) are summarized to find a "batch level" fix that should be applied (an LLM takes this batch and generates the batch level fix), which is then added back to the original prompt & the iteration continues.

**Strengths:**

I like the idea of using intermediate feedback instead of depending upon just the final feedback. (Specifically, the batch level fix instead of per-instance fix).
The paper is well written and covers good relevant literature.

**Weaknesses:**

What are the situations where intermediate feedback is easily / readily available and can be as useful as having access to a final critic?

The major weakness of this paper is :
1. Choice of dataset : They choose only two datasets TravelPlanning & Guan et al to showcase their results. I would expect results on a larger number of datasets to establish the claims.
2. The results on these datasets is not encouraging. Specifically, in Guan et al, the they only improve slightly on Household domain. Moreover, the Final pass rate on TravelPlanning domain doesn't move the needle by much as well. For instance, [recent work](https://arxiv.org/abs/2405.20625) uses a bank of critics (similar to intermediate feedback) and achieves over 20% final pass rate - compared to 3.89%.

The results on the Travel planning domain highlights my concern as well, on the belief that intermediate feedback can replace a final critic. Dependance of these individual feedback pieces can be quite complex and the paper relies on "batch fix"  to identify it, which doesn't seem to be scalable / may require lots of iterations. For instance, in TravelPlanning changing attributes like budget, accomodation preferences etc. can impact host of other changes which would be harder to capture by just looking at the intermediate feedbacks.

Finally, while I agree if we know the "impactiful parts" of the input, given a feedback we can update those impactful parts - this may not apply to the author's example in ReAct. For instance, some works highlight the futility of "thoughts" in ReAct style prompting and that several parts of the prompt can infact be ignored / used which are not aligned with how human's infer the prompt. Therefore, depending upon an LLM may not be robust in identifying these "impactful parts" and some explainable AI strategy maybe better suited.

Minor :
Can the authors point me to details on their "summarizing team", how was the data collected from them (beyond high level description in the main paper) and study / IRB details.

**Questions:**

See Weaknesses.

---

> ### Author Response · Authors · 2024-11-16
> **Rebuttal**
>
> W1. Because our paper is set in a unique context where the solution checker is either unavailable (as in TravelPlanner) or very expensive (as in PDDL generation) during the generation/training process of an LLM agent task, the two datasets were selected accordingly. At the time of our submission, we did not find any other publicly available datasets that fit this scenario. However, we believe the current two models already cover two categories of tasks that align with this setting: 1) The PDDL generation task provides precise but costly feedback, testing the exact translation abilities of LLMs—a crucial skill for generating correct code; 2) In contrast, the TravelPlanner environment, using Reflexion, provides low-cost but less accurate feedback without access to ground-truth information. Furthermore, due to the high cost of running LLM-agent tasks, many well-regarded papers, such as ReAct, also test only on two datasets, as we do in our paper.
>
>
> W2. We do not think our improvement scale on the datasets is unsatisfactory. For Guan et al. (the PDDL generation task), we have reduced approximately half of the total number of errors, even in the household domain you mentioned. The remaining errors, as discussed in Section 5 of our paper, are highly case-specific and theoretically challenging to resolve at the prompt level. For the TravelPlanner benchmark, we need to emphasize that the recent work you referred to is one of the studies that explicitly hacks the checker during the generation phase. They use external code executors to facilitate the generation process, which completely undermines the original purpose of the dataset, as clearly stated in the TravelPlanner repository: https://github.com/OSU-NLP-Group/TravelPlanner?tab=readme-ov-file#%EF%B8%8Fwarnings. In this case, their improvement scale is entirely incomparable to works like ours that adhere to the requirements.
>
>
>
>
> > The results on the Travel planning domain highlights my concern as well, on the belief that intermediate feedback can replace a final critic.
>
>
> Our method, REPROMPT,  uses the knowledge fetched in self-reflection phased into the prompt engineering phase. On the one hand, this will avoid early-stage errors, and more importantly, **this will reduce the randomness caused by ReAct and Reflexion where some errors might not receive good enough feedback in a specific scenario**. By summarizing the feedback, REPROMPT ensures that all common problems, such as balancing a budget, are highlighted in the prompt, leaving the tasks of ReAct and Reflexion to address more case-specific issues, such as a budget constraint caused by the significantly high price of hotels in a certain city on a specific day. We invite you to review the converged prompt provided in the appendix to see how constraints become more explicitly addressed after prompt optimization.
>
>
> > Therefore, depending upon an LLM may not be robust in identifying these "impactful parts" and some explainable AI strategy maybe better suited.
>
>
> We agree with you that, given the current capabilities of LLMs, they may not reliably identify these “impactful parts.” However, similar to many other AI/ML research areas, our work focuses on finding ways to automatically solve problems and reduce the human labor involved. On a broader level, the entire domain of Automatic Prompt Engineering (APE) aims to replace manual prompt engineering, even though APE is not necessarily superior yet. We believe that, as one of the works in this pioneering research direction, our study can serve as a step toward achieving a point where automatic processes outperform human efforts.
>
>
> >  Can the authors point me to details on their "summarizing team"
>
>
> We apologize for the confusion. We use the word “team” in this context—its only occurrence in the paper—to differentiate it from the other components of the optimizer: the actor and the feedback generator. Therefore, this part is still performed by an LLM, which is fully automated and does not require IRB approval. For your reference, the prompt for the summarizer is provided in the appendix, and we have also discussed how summarizers are designed in our method section. We have updated our paper to remove the potential confusion here.

---

### Official Review · Reviewer_FS8P · 2024-11-05

**Soundness:** 2
**Presentation:** 2
**Contribution:** 2
**Rating:** 5
**Confidence:** 4

**Summary:**

The paper focuses on the problem of prompt optimization for LLMs for tasks in which the final evaluation of the task performance is either difficult or not possible to access. The proposed method aims to optimize step-by-step instruction following prompts (i.e., chain-of-thought) by gathering feedback (either human or LLM-generated) on the intermediate steps in the LLM-generated response, summarizing the key points of feedback, and using an LLM to generate an updated prompt that incorporates the single most important point of feedback for that iteration of the method. This process repeats until some notion of convergence has been satisfied. The method, RePrompt, is experimentally evaluated on two tasks: (1) PDDL action precondition generation and (2) constraint-satisfying travel planning.

**Strengths:**

I think the paper identifies a challenging problem with prompt optimization for black-box LLMs. Many existing methods rely on task success signals to improve the associated prompt, but such a signal may be expensive or not available. Tackling the problem by utilizing feedback about the LLM's intermediate steps is a promising research direction.

**Weaknesses:**

There are several weaknesses of note. I first list the weaknesses, then provide a clear statement of my recommendation along with arguments that support the recommendation, concluding with some suggestions about how to change my decision. Finally, there are a few general comments that did not impact my review recommendation.

Weakness:
- The paper is rough to read, making it generally difficult to understand the points being made.
- The description of the method is difficult to follow and is not formalized mathematically or algorithmically.
- To the best of my understanding, several of the stated contributions are not novel and appear in existing work.

I recommend rejection primarily for the following reasons.

1. The paper is difficult to read, making it hard to follow both the description of the method and the experimental design. For example, PromptAgent is used as a baseline for the Travel Planner experiment, but it is not described at all. I do not enumerate all of the comprehension challenges here; it is generally pervasive.

2. Several of the claimed contributions are not novel. Below, I go through these individually.

2A. The first claimed contributed is to propose using a "gradient-based"-like prompt optimization. First, if this is a key contribution, it is necessary to make explicit what the gradient analog is; is the idea that the feedback provides something like a semantic direction in which to step to improve the prompt? Second, there are several other works that propose an analogy to gradient-based optimization for black-box LLMs [1,2], one of which is cited in this paper.

2B. The second claimed contribution is a summarization-based method for providing specific feedback. There is existing work that also explicitly includes a summarizer over batches of feedback for prompt optimization [3].

2C. The third claimed contribution is that the method does not require a solution checker. It is unclear to me how this is different than ReAct and Reflexion, both of which are described in this paper.


[1] Yuksekgonul, Mert, et al. "TextGrad: Automatic" Differentiation" via Text." arXiv preprint arXiv:2406.07496 (2024).
[2] Pryzant, Reid, et al. "Automatic prompt optimization with" gradient descent" and beam search." arXiv preprint arXiv:2305.03495 (2023).
[3] Chen, Yongchao, et al. "Prompt optimization in multi-step tasks (promst): Integrating human feedback and preference alignment." arXiv preprint arXiv:2402.08702 (2024).

My recommendation is based on my understanding of the paper. As it stands, I think the main possible way my recommendation could be improved is if I have significantly misunderstood the method and contributions, which is entirely possible since I found the paper confusingly written and unclear. I would recommend that the authors significantly rework Section 3.2 by introducing unabiguous notation, providing a concrete example to walk through with their method that utilizes the new notation, and clearly identifying the novel steps that differentiate their approach from ReAct/Reflexion.


Additional Comments (these did not impact my recommendation):
- I recommend that the authors clearly explain what makes an LLM-based system agentic. This would fit well in the introduction.
- I think an algorithm block would be one good format to present the method that should significantly reduce ambiguity.
- PDDL is an acronym for Planning Domain Definition Language (the paper writes "Planning Definition and Domain Language).
- The subsections in Section 5 appear to be LaTex subsubsections.

**Questions:**

1. What are the conditions under which the updated prompt converges? In the steps on lines 209-219, does convergence mean that all proposed solutions are already included? Is the feedback empty? Clarifying how this iteration terminates is important.

2. Line 226 refers to manually encoded solutions in the optimizer. What are these common problems and why is it better to manually encode solutions rather than allow the method to address them?

3. Line 243 refers to in-context learning, which I think is the first and only place this is referenced. Further, line 293 refers to zero-shot generation. How is in-context learning actually used in the experiments?

---

> ### Author Response · Authors · 2024-11-16
> **Rebuttal**
>
> Thank you for your review. Here, we address your concerns one-by-one.
>
>
> # Weakness
>
>
> 1. Thank you for pointing this out. We apologize for any confusion caused by our previous version, and we have updated the paper accordingly. Unfortunately, due to page limitations and the nature of LLM agents requiring multiple iterations—further multiplied in batch settings like REPROMPT, which can lead to hundreds of feedback rounds—it is not feasible to include even a single example that illustrates each part of our algorithm. Additionally, due to the page limit, we find it impossible to include the prompts in the main paper, which we understand is a vital part of LLM-based papers. We invite you to review our updated paper, which we believe will help clarify your questions.
>
>
> 2. We agree that all the related work you mentioned is relevant, as our work lies at the intersection of self-reflection and prompt engineering. However, we believe the unique domain of our work also differentiates it from previous work. Here is our detailed comparison, highlighted in bold, for each point you raised.
>
>
> 2A. Indeed, there have been some works, if not many, that use a similar idea of ``gradient-based” method for prompt optimization for LLMs, however, none of these works are in the domain of LLM ** agents**, and consider the ReAct/Reflexion framework that is essential in getting a good performance in the domain.
>
>
> 2B. Thanks for pointing out this related work. We have added this work to our paper. Our paper differs from the existing method both in the fact that **we are considering ReAct in the loop, and the detailed way of how the summarizer is constructed and used by the optimizer**. More specifically, compared to PromSt, their method focuses on human-provided feedback, which is much more accurate than ours. In contrast, our optimizer can support less informative feedback as well.
>
>
> 2C. Our method REPROMPT **uses the knowledge fetched in the self-reflection phase within the prompt engineering phase via summarization**. This avoids early-stage errors, and more importantly, reduces the randomness caused by ReAct and Reflexion, where some errors might not receive adequate feedback in a specific scenario. By summarizing the feedback, RePrompt ensures that all common problems, like balancing budgets, are highlighted in the prompt while leaving the job of ReAct and Reflexion to address more case-specific issues, such as budget constraints caused by the significantly high price of hotels in a certain city on a specific day.
>
>
> # Questions
>
>
> 1. Convergence means the prompt does not change after another round of updates. Similar to convergence in standard ML model training, the converged prompt does not necessarily lead to perfect performance in all cases and could still have feedback. However, with this remaining feedback, our optimizer will determine whether the summarized feedback is already included in the current prompt or is case-specific, and it will ignore this feedback accordingly.
>
> 2. We have provided the optimizer prompt in our appendix, and the three common questions are explicitly included there. We chose to include them in the prompt because we found these are problems that frequently appear. Optimizers do not always solve these common problems in one trial and may waste many iterations before arriving at similar solutions. To reduce the cost and improve the process's efficiency, we chose to include them in the optimizer prompt. However, theoretically, not adding them does not change the final performance or conclusion.
>
> 3. Sorry for the confusion. We have updated the related paragraph in the new version. To directly answer the question: we provide REPROMPT and the low-level solver with examples of the task, but not in the exact scenario they are facing. We do allow in-context learning (i.e., examples in the prompt) in the initial prompt, and as discussed in the paragraph of original line 243, the corresponding paragraph remains unchanged. In the PDDL generation task, we generate the response only once without previous trials in each specific scenario.

---

> > ### Comment · Reviewer_FS8P · 2024-11-24
> >
> > Thank you for your response.
> >
> > Regarding my criticisms of novelty that points to existing works, I agree with the following:
> > - this is the first work, to the best of my knowledge, that specifically uses "gradient-based" prompt optimization for *LLM agents*, as opposed to non-agentic LLM tasks; however, I think this distinction is minor.
> >
> > - the summarizer LLM used in REPROMPT is different in some details than the cited work; however, I still think the problem being addressed by this step (i.e., distilling a large amount of feedback to be efficiently incorporated during optimization) is very similar and thus the contribution provides little insight.
> >
> > Further, thank you for further clarifying how REPROMPT is different than ReAct/Reflexion with respect to requiring a solution checker.
> >
> > Regarding my questions, I find the answers to all be satisfying.
> >
> > My criticism about the readability and a lack for any formalization remains. The updates to the paper do help clarify some aspects of the method.
> >
> > This response has addressed some of my concerns about novelty, but I remain critical of the significance of some of the contributions. Clarity of the method has been marginally improved with the additional content. I will raise my score to reflect these points.

---

> > > ### Author Response · Authors · 2024-11-24
> > > **Further Response by Authors**
> > >
> > > Thank you for your response. We are happy to see that you are satisfied with our previous response to your questions, and here we further respond to the remaining concerns you kept:
> > >
> > > > however, I still think the problem being addressed by this step (i.e., distilling a large amount of feedback to be efficiently incorporated during optimization) is very similar and thus the contribution provides little insight.
> > >
> > > We argue that while summarizers may share similarities, their utilization introduces critical distinctions. For instance, PROMST employs a summarization approach driven by feedback on failure cases, using an iterative process to diagnose errors made by the task-planning LLM agent and refine prompt descriptions to preempt such mistakes. In contrast, our method, REPROMPT, summarizes all sampled trajectories, thereby eliminating the need for a solution checker and directly optimizing specific steps within the multi-step guidelines provided in the prompt. We emphasize that addressing similar high-level challenges with diverse designs can yield complementary insights and make significant contributions to the community. This is exemplified by comparisons such as TRPO [1] versus PPO [2], QMIX [3] versus MADDPG [4], and AdamW [5] versus RMSProp [6].
> > >
> > > > this is the first work, to the best of my knowledge, that specifically uses "gradient-based" prompt optimization for LLM agents, as opposed to non-agentic LLM tasks; however, I think this distinction is minor.
> > >
> > >
> > > The differences between LLMs and LLM agents primarily stem from two aspects. First, tasks for LLM agents are generally more complex than those for pure LLMs. Consequently, many widely used methods, such as CoT [7], ReAct [8], and Reflexion [9], have been developed to significantly enhance performance. Therefore, when performing prompt optimization for LLM agents, it is crucial to build upon these architectures and ensure that the proposed methods are compatible with these established techniques.
> > >
> > > Second, tasks for LLM agents often have stricter format requirements. For instance, in the TravelPlanner benchmark, adhering to specific formats is necessary to utilize functions like CostEnquiry. If an automatic prompt engineering algorithm is allowed to make arbitrary changes, as is often the case in general LLM tasks, it could violate these format constraints and fail to optimize prompt performance effectively.
> > >
> > > These two factors combined make automatic prompt engineering for LLM agents a unique challenge, where conclusions drawn from general LLM tasks do not easily generalize. For example, as shown in Table 2 of our paper, while PromptAgent is one of the most effective automatic prompting algorithms for general LLM tasks, it fails to outperform the baseline Reflexion in LLM agent tasks.
> > >
> > > > My criticism about the readability and a lack for any formalization remains.
> > >
> > > We are sorry to hear that. While we strive to make our paper accessible and easy to read for all, we would greatly appreciate it if you could provide more details about which aspects of our algorithm you still find confusing. Is it the general workflow, the summarizer, or the optimizer? We will do our best to address your concerns and improve the clarity.
> > >
> > > Thank you again for your response, and we hope our further response can help address your concerns. Please let us know if you have any further questions.
> > >
> > > [1]. Schulman J. Trust Region Policy Optimization[J]. arXiv preprint arXiv:1502.05477, 2015.
> > >
> > > [2]. Schulman J, Wolski F, Dhariwal P, et al. Proximal policy optimization algorithms[J]. arXiv preprint arXiv:1707.06347, 2017.
> > >
> > > [3]. Rashid T, Samvelyan M, De Witt C S, et al. Monotonic value function factorisation for deep multi-agent reinforcement learning[J]. Journal of Machine Learning Research, 2020, 21(178): 1-51.
> > >
> > > [4]. Lowe R, Wu Y I, Tamar A, et al. Multi-agent actor-critic for mixed cooperative-competitive environments[J]. Advances in neural information processing systems, 2017, 30.
> > >
> > > [5]. Loshchilov I. Decoupled weight decay regularization[J]. arXiv preprint arXiv:1711.05101, 2017.
> > >
> > > [6]. Graves A. Generating sequences with recurrent neural networks[J]. arXiv preprint arXiv:1308.0850, 2013.
> > >
> > > [7]. Wei J, Wang X, Schuurmans D, et al. Chain-of-thought prompting elicits reasoning in large language models[J]. Advances in neural information processing systems, 2022, 35: 24824-24837.
> > >
> > > [8]. Yao S, Zhao J, Yu D, et al. ReAct: Synergizing Reasoning and Acting in Language Models[C]//The Eleventh International Conference on Learning Representations.
> > >
> > > [9]. Shinn N, Cassano F, Gopinath A, et al. Reflexion: Language agents with verbal reinforcement learning[J]. Advances in Neural Information Processing Systems, 2024, 36.

---

### Meta-Review · Area_Chair_JjPc · 2024-12-19

**Metareview:**

**Summary**: The paper introduces RePrompt, a method for automatic prompt engineering in LLM agents. Unlike existing methods that rely on final solution checkers, RePrompt leverages intermediate feedback during multi-step tasks to iteratively optimize prompts through summarization and reflection. The method is tested on two reasoning benchmarks—PDDL generation and Travel Planning—showing modest improvements over baseline methods like Reflexion and PromptAgent. The proposed approach emphasizes accessibility and adaptability to scenarios where final solution checkers are unavailable or prohibitively expensive.

**Strengths**:
- Important and challenging problem focus: Reviewer `FS8P` highlights that the paper addresses the challenging problem of prompt optimization without relying on final evaluators, which is particularly relevant for real-world LLM agents where intermediate feedback is often more practical and accessible.
- Decent conceptual contributions: The reviewers acknowledge that the approach adapts concepts like "gradient-based" optimization to prompt engineering, tailored for multi-step reasoning tasks in LLM agents. This distinction from standard APE methods was clarified in the rebuttal, emphasizing the unique requirements of LLM agents.

**Weaknesses**:
- Limited dataset and benchmarks: Reviewer `vpmw` criticizes the narrow scope of evaluation, which includes only two datasets—PDDL generation and Travel Planning. This limits the generalizability of claims about RePrompt’s effectiveness.
- Modest experimental gains: Reviewers express concerns about the scale of improvement. For instance, Reviewer `vpmw` notes that the final pass rate improvement in Travel Planning is marginal, and performance on PDDL generation improves only in specific domains like "Household".
- Lack of novelty: Most reviewers question the novelty of RePrompt’s contributions compared to prior works like Reflexion, PromptAgent, and PromSt. While the rebuttal clarified distinctions, Reviewer `FS8P` remains unconvinced about the broader significance of the claimed contributions.
- Unclear, difficult to follow writing: Reviewer `FS8P` finds the paper difficult to follow due to a lack of formalization and algorithmic detail, particularly in Section 3.2. The rebuttal provided clarifications, but the fundamental clarity of the method remains a weakness.

**Recommendation**: This paper presents a timely contribution to the field of prompt optimization for LLM agents. However, limitations in dataset scope, modest experimental gains, novelty of contributions, and clarity reduce its overall impact. While the reviewers appreciated the rebuttal and noted incremental improvements, the concerns about generalizability and presentation persist. As such I vote to Reject this paper.

**Additional Comments On Reviewer Discussion:**

The authors addressed several key concerns during the rebuttal:

- Novelty/Contributions over prior work: The rebuttal clarified how RePrompt builds on prior methods like Reflexion and PromptAgent while focusing on intermediate feedback optimization for LLM agents. Reviewer `FS8P` acknowledged the distinction but viewed it as a minor contribution.
- Dataset scope: The authors explained that the chosen datasets reflect scenarios where solution checkers are impractical, aligning with the paper's goals. However, the narrow dataset scope remains a limitation.
- Writing clarity: The authors added algorithmic details and provided examples in the appendix to improve readability, which partially addressed Reviewer `FS8P`'s concerns.
- Experimental results and gains: The authors defended the modest improvements by emphasizing the complexity of LLM-agent tasks and the inherent challenges of prompt optimization. Reviewer `vpmw` appreciated this clarification but noted that the results still fell short of expectations.

Following the rebuttal, Reviewers `FS8P` and `TG95` raised their scores slightly but maintained reservations about the paper’s clarity and significance. Reviewer `vpmw` remained critical of the experimental scope and results. Overall, the average score is below the acceptance threshold, so this paper is a clear Reject.

---

### Decision · Program_Chairs · 2025-01-22

Reject